# Remote heart rate monitoring - Assessment of the Facereader rPPg by Noldus

**Simone Benedetto**[1]*, **Christian Caldato**[1], **Darren C. Greenwood**[2,3], **Nicola Bartoli**[1], **Virginia Pensabene**[4,5], **Paolo Actis**[4]

**1** TSW XP Lab, Via Terraglio, Treviso, Italy, **2** Leeds Institute for Cardiovascular and Metabolic Medicine, University of Leeds, Leeds, United Kingdom, **3** Leeds Institute for Data Analytics, University of Leeds, Leeds, United Kingdom, **4** School of Electronic and Electrical Engineering, University of Leeds, Leeds, West Yorkshire, United Kingdom, **5** School of Medicine, Leeds Institute of Biomedical and Clinical Sciences, University of Leeds, Leeds, West Yorkshire, United Kingdom

* benedetto.simone@gmail.com

**Data Availability Statement:** All relevant data are within the manuscript and its Supporting Information files.

## Abstract

Remote photoplethysmography (rPPG) allows contactless monitoring of human cardiac activity through a video camera. In this study, we assessed the accuracy and precision for heart rate measurements of the only consumer product available on the market, namely the Facereader™ rPPG by Noldus, with respect to a gold standard electrocardiograph. Twenty-four healthy participants were asked to sit in front of a computer screen and alternate two periods of rest with two stress tests (i.e. Go/No-Go task), while their heart rate was simultaneously acquired for 20 minutes using the ECG criterion measure and the Facereader™ rPPG. Results show that the Facereader™ rPPG tends to overestimate lower heart rates and underestimate higher heart rates compared to the ECG. The Facereader™ rPPG revealed a mean bias of 9.8 bpm, the 95% limits of agreement (LoA) ranged from almost -30 up to +50 bpm. These results suggest that whilst the rPPG Facereader™ technology has potential for contactless heart rate monitoring, its predictions are inaccurate for higher heart rates, with unacceptable precision across the entire range, rendering its estimates unreliable for monitoring individuals.

## Introduction

There is a growing interest in technologies related to the recording and monitoring of personal health parameters. In the current literature there is not yet a general agreement on the definition of personal health monitoring, which includes telecare, assistive technologies, environmental intelligence and wearable health sensors [1]. A review on the subject suggests that "monitoring of personal health" refers to any electronic device or system that monitors a health-related aspect of a person's life outside a traditional clinical or hospital setting. Examples include GPS tracking devices used with patients with mental disorders, blood pressure monitors and smart clothes capable of measuring physiological parameters [2, 3, 4, 5]. Personal health recording systems are more than just static patient data containers; they combine data, knowledge, tools and software, which help both patients with identified needs and generic consumers to become active participants in their health care [6]. Health monitoring

**Funding:** This work was supported by TSW XP Lab, which only provided financial support in the form of authors' salaries [SB, CC, NB] and/or research materials. The funder had no role in study design, data collection and analysis, decision to publish, or preparation of the manuscript.

**Competing interests:** The authors declare that the funding organization (TSW XP Lab) only provided financial support in the form of authors' salaries [SB, CC, NB] and/or research materials, and did not play a role in the study design, data collection and analysis, decision to publish, or preparation of the manuscript. The specific roles of these authors are articulated in the "Author Contribution" section. The authors also confirm that this commercial affiliation does not alter their adherence to all PLOS ONE policies on sharing data and materials.

technologies are currently being developed for a multitude of customers of all ages and health conditions aiming to integrate medical care environments with health monitoring outside traditional settings [1]. A primary driving factor is the rapid ageing of the population, which is expected to heavily impact on the performance of health systems in many countries, potentially exceeding the available medical resources [7]. Patients, policymakers, providers, tax-payers, employers, and other stakeholders have increasing interest in using personal health records to reduce healthcare costs, without affecting the quality and the efficiency of the healthcare delivery [8].

The monitoring of physiological information is very important for assessing health and access to physiological data is not only necessary in clinical setting but it is becoming increasingly so also in other environments and applications related, for example, to telemedicine [9–13], personal fitness [14–17], e-commerce [18], trading [19, 20] and mental stress caused by the interaction with technology [21–26]. Accurate and precise self-monitoring devices therefore provide potential benefits both to the individual user, by providing real-time feedback on specific physiological parameters, to the health care providers and also to those involved in retail intelligence and analytics [27, 28]. For example, typical physiological and neuroscience research techniques used to study cognitive and affective processes of individuals such as electroencephalography (EEG), functional magnetic resonance imaging (fMRI), eye tracking, biometrics of heart rate, galvanic skin response, and facial expression recognition, are becoming increasingly popular consumer neuroscience methods [29]. These techniques contribute to a deeper understanding of consumers behaviours by gathering quantitative information on their physical and mental state [30]. A growing interest has also raised in the context of emotion detection and recognition, where several devices are now available (e.g. Affectiva, Emotient—An Apple Company, Eyeris, Kairos Ar. Inc., Noldus, nViso, Realeyes) [31].

The conventional and well-established methods to capture physiological information, like electrocardiogram (ECG) or photoplethysmogram (PPG), require the application of electrodes or transducers on the skin (e.g. wet adhesive Ag/AgCl electrodes) during the monitoring period. These methods, although non-invasive, are bothersome, and perhaps irritating and distracting.

Recently, there has been an increasing interest in alternative and less obtrusive methods for monitoring physiological information such as, laser doppler velocimetry for measuring red blood cell velocity, electromagnetic approaches for heart and respiration monitoring, microwave systems or ultrasonic proximity sensor for respiration detection [32, 33].

The remote-photoplethysmography (rPPG) is a low-cost, non-contact and pervasive technique for measuring heart rate (HR) and to infer other psychophysiological data including heart rate variability, respiration rate, blood pressure and oxygenation [34, 35], quality of sleep, heart rhythm disturbances [36], and also mental stress [37] and drowsiness [38]. Its ease of use, low cost and convenience make it an attractive method for biomedical and clinical research as it allows remote heart rate measurements with a simple camera or a smartphone and it can also be integrated with augmented reality platforms [39]. The information acquired through the rPPG essentially refers to the cardiovascular functioning: the periodic blood flow and therefore the variations of blood volume in tissues that follow each cardiac cycle affects the optical properties of the tissues allowing those who are using this technology to measure HR remotely. For this reason, the reflection of the light that can be observed on the regions of the facial skin. This reflection of light is influenced not only by the various phenomena of interaction between light and skin, but also by the change in the volume of blood and the movement of the wall of blood vessels [40, 41]. Based on this principle, an accurate measurement of these changes generates a plethysmographic signal. Research has shown that, given suitable illumination, ambient light can be sufficient to obtain a plethysmographic signal [42] from changes in light reflected from facial skin and thus it is possible to measure and infer on the

physiological phenomena of interest. The only hardware required to perform rPPG imaging is a standard camera. Although several techniques based on the use of infra-red (IR) or near infra-red (NIR) cameras exist [43, 44], the most developed and employed algorithms use a colour model method based on red, green and blue (RGB) imaging to acquire a signal from a distance of up to several meters [45, 46, 47]. In technical terms, the main difference between these two methods lies in the fact that both IR and NIR cameras allow a more accurate estimate of HR parameters and exploit the information provided by blood volume variation of vessels. In turn, RGB camera-based method (green light channel), does not provide such a profound and focused estimate of HR, and consider a wider and less focused range of processes which influence the optical properties of the tissues [48]. According to Wang and colleagues [49], the RGB camera-based method presents two main limitations: it is difficult to accurately estimate HR under low-light conditions and under significant ambient light fluctuations; these last two factors together with the head and body movements can drastically affect rPPG signal detection by generating strong artefacts. The general recommendation for proper measurement of HR is to keep the illumination constant and restrict individual movements.

The described methods capture the subject's face on a video from which the plethysmographic signal is recovered using several image processing techniques and transformations. The rPPG technology, from its first presentation in 2007 [50], has been studied and developed to demonstrate its feasibility first in controlled environments and conditions, and then in increasingly realistic conditions and scenarios. Research has shown that reliable HR measurement can be achieved using low-cost, consumer-grade digital cameras and ambient light sources. Current literature on rPPG focuses on improvement over existing methodology by considering those imaging acquisition factors (environmental lighting, subject movement, and image sensor spectrum sensitivity) that, at this time, represent the main limitation to an optimal rPPG measurement and therefore to collect accurate physiological data (e.g. [51, 52]). All these methods are of interest for the easy, convenient and large-scale deployment of the non-contact HR monitoring technologies.

In this respect the goal of this study was to critically assess the accuracy of a consumer rPPG system by Noldus with respect to HR monitoring and compared its performance to a gold standard electrocardiograph. The Facereader[TM] rPPG system by Noldus monitors HR activity through a patented rPPG technology [53]. Although on the market there are probably alternative and more advanced technologies such as the Vital Signs Camera by Philips, which is available for licensing to third part manufacturers, to the authors' knowledge, the rPPG by Noldus is the only consumer product available on the market and up until today just one study involving this specific tool has been carried out [54]. Although recent evidence suggests that reliable HR measurement can be achieved using different rPPG algorithms [25, 49, 51, 55, 56], the need of further validations and cross comparisons is crucial. The objective of this study is to contribute to the improvement of this kind of technology, which has the potential to assess and monitoring the personal psychophysical status in a simple, convenient and non-invasive way with important applications (e.g. consumer analysis, e-commerce, personal fitness, driving conditions, telemedicine, customer neuroscience) and therefore to its diffusion. Our validation, in case of significant results, could assume a strong relevance regarding the potential but realistic application of remote heart rate monitoring in workplace environments.

## Materials and methods

The accuracy and precision of the Facereader rPPG by Noldus (Noldus Information Technology bv, Wageningen—The Netherlands) for measuring HR was assessed with respect to an ECG criterion measure. The ProComp Infiniti T7500M (Thought Technology LTD, Toronto,

Canada), is a professional 8 channel multi-modality encoder for real-time, computerized bio-feedback and data acquisition used in the clinical and experimental field and constitutes a gold-standard for the measurement of physiological signals. For ECG recorded, the electrode placement sites were prepared by standardized procedures of cleaning, shaving, and abrading the skin to improve signal acquisition and to minimize noise artefacts. Three silver/silver-chloride self-adhesive electrodes were placed in proximal position on the upper torso following the second standard deviation according to the Einthoven triangle [57]. On Einthoven's triangle, the theory of unipolar electrocardiographic leads, and the interpretation of the precordial electrocardiogram). ECG data were recorder and processed using BioGraph Infinity (Thought Technology LTD, Toronto, Canada). HR data was converted to beats per minute (bpm) automatically by the data acquisition software program prior to analysis. For rPPG, the facial landmark estimation is achieved using the Active Appearance Modelling (AAM) technique [54, 58] that has been improved and integrated in the Facereader framework [59]. The AAM is a method of matching statistical models of appearance to images that consists in an efficient iterative matching algorithm by learning the relationship between perturbations in the model parameters and the induced image errors [54, 58]. Using the selected facial regions, skin colour changes are tracked for observing the periodic components caused by the blood volume changes at each heartbeat [60].

The rPPG system requires the use of a video acquisition source for recording the face of the participants. To this end a Logitech HD Pro Webcam C920–1080 HD was employed. The resolution of the videos was 1280 x 720 pixel, and the frame-rate acquisition was 30 fps for a duration of 20 minutes. In order to ensure a high quality of rPPG signal acquired from the skin surface, the whole experiment was carried out under constant lighting conditions. Both the ambient (illumination) and the screen (luminance) were controlled during the entire experiment. These parameters were assessed by an Extech 403125 digital light meter (Extech Instruments, Nashua, NH) pointed towards the screen and placed 5cm above participants' head and laterally centred with respect to their head. Overall, the total amount of light impacting on participants' face coming from both the ambient lighting and the screen was kept constant during the entire experiment and did not vary because of the stimuli presentation triggered by the Go/No-Go task. The distance between participants and the 24" LCD stimulus screen (Dell P2414H; www.dell.com) was approximately 60 cm (see Fig 1).

Twenty-Four healthy participants (11 females) of Western European descent took part in the experiment. The selection of such a specific population was mainly due to the fact that we wanted to reduce the possible effects of skin tone, which constitutes one of the main detection issues for rPPG. All participants gave written informed consent before participation. We excluded participants with cardiovascular diseases (CVD) and with neurological or cognitive disorders. The study was performed in a controlled experiment room at TSW XP Lab, Treviso—Italy (www.tsw.it) complying with the Declaration of Helsinki. The TSW XP Lab Ethics Committee approved the study. Participants were asked to sit in front of the computer screen and alternate two periods of rest with two stress tests (Go/No-Go task), as follows:

- Rest [5 min]

- Stress test (Go/No-Go task) [5 min]

- Rest [5 min]

- Stress test (Go/No-Go task) [5 min]

The Go/No-Go task required participants to press the spacebar when they saw a green rectangle appeared (Go) but refrain from pressing the spacebar when they saw a blue rectangle

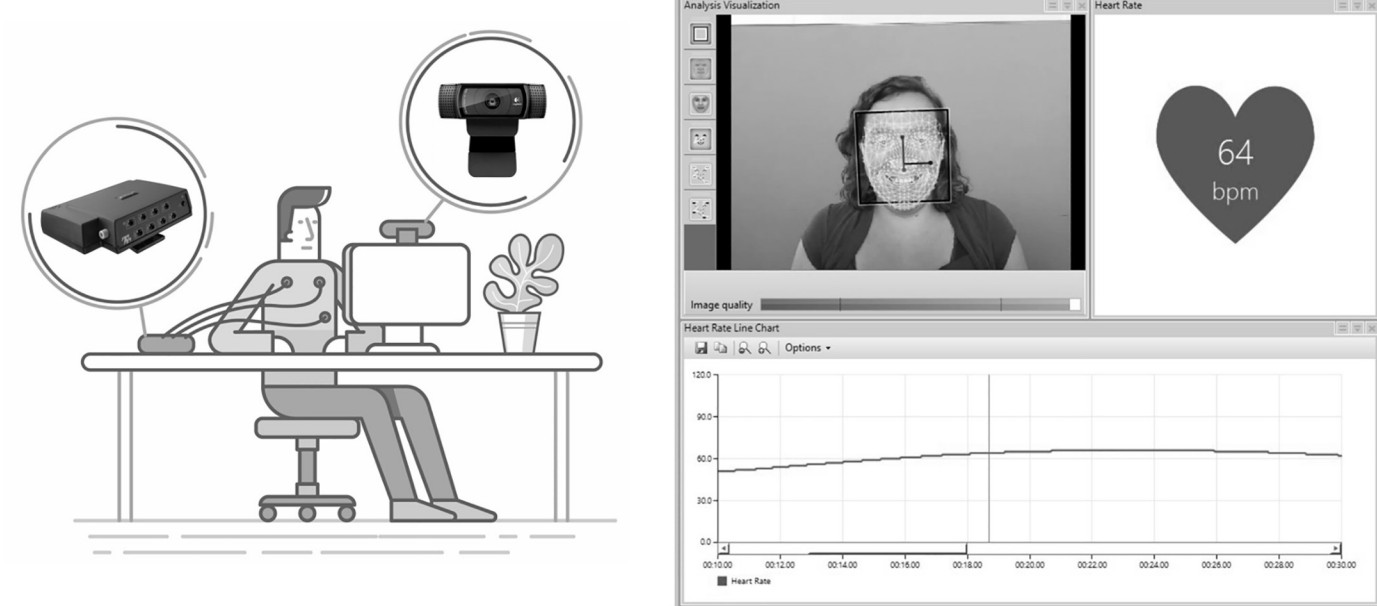

**Fig 1. Experimental setup and software.** On the left, a schematic representation of the experimental setup and the devices: ProComp Infiniti T7500M (ECG recording) and Logitech HD Pro Webcam (video acquisition). On the right, a screenshot of the Facereader rPPG software by Noldus.

(No-Go). The blue and green rectangles could be either vertically or horizontally aligned. The vertical rectangle had a higher probability of being green (Go trial) and the horizontal rectangle had a higher probability of being blue (No-Go trial). Participants got information about the orientation of the rectangle (cue) shortly before the colour of the rectangle was revealed [61].

The goal of the experiment was to collect enough HR data spanning a range of BPMs as wide as possible. HR was simultaneously acquired for 20 minutes using the ECG criterion measure and the Facereader rPPG by Noldus. With the aim of improving the overall data quality, the rPPG analysis was carried out offline. Given that Facereader allows collecting data at 8 Hz and that BioGraph Infinity allows pre-processing at 8 Hz, we decided to use this frequency for processing cardiac data. Moreover, since movements from the participant may cause artefacts in the HR monitoring, we have manually removed any motion-induced artefacts. Agreement between the Facereader rPPG and the ECG gold standard was estimated using the Bland-Altman method, adapted to consider repeated measures from the same person when the true value varies over time [62, 63]. Bland-Altman plots are widely used to evaluate the agreement among two different instruments or two measurements techniques. This provided an estimate of agreement between the rPPG and ECG in the instantaneous value of the changing heart rate. We also estimated the intraclass correlation coefficient (ICC), which indicates how strongly units in the same group resemble each other, as a complementary measure of agreement. All statistical analyses were performed using StataCorp Stata 15.1.

## Results

Table 1 shows means, standard deviations (SD) and ranges for age, weight, height, and Body Mass Index (BMI) of the participants.

The dataset consisted of 230400 samples of data (1200 seconds x 24 participants x 8 Hz). However, since both the ECG and the Facereader rPPG produced several disruptions to continuous HR detection, the dataset was reduced by around 23%. The final dataset was then made of 177629 samples. Fig 2 shows all time-synced ECG and Facereader rPPG ordered by

**Table 1. Means, standard deviations and ranges for Age, Weight, Height, and BMI.**

| Variables | Male | | Female | |
|---|---|---|---|---|
| | Mean ± SD | Range | Mean ± SD | Range |
| Age (Years) | 31 ± 5 | 23–38 | 27 ±3 | 23–31 |
| Weight (kg) | 76 ± 10 | 62–92 | 59 ± 8 | 45–71 |
| Height (cm) | 178 ± 5 | 168–189 | 166 ± 3 | 160–174 |
| BMI (kg/m$^2$) | 24 ± 2 | 20–27 | 21 ± 3 | 16–25 |

ECG data in aggregate, with the rPPG estimate demonstrating wide variability and lack of responsiveness to changing heart rate recorded by ECG.

The Facereader rPPG revealed a mean bias of 9.8 bpm (95% CI—Confidence Interval: 9.7 to 9.9 bpm). As to the limits of agreement (LoA) between the Facereader rPPG and criterion measure the upper LoA was 46 bpm, whereas the lower LoA was -26 bpm (Fig 3). The ICC between Facereader rPPG and gold standard ECG was 0.75 (95% CI: 0.64 to 0.86).

Furthermore, the extent of agreement varied substantially across the range of heart rates (see Fig 4). The Facereader rPPG tends to overestimate lower heart rates ($< 80$ bpm) compared to the ECG and underestimates higher heart rates ($> 80$ bpm) compared to the ECG. Since the previous Bland-Altman Plot (see Fig 3) and relative statistics related to the mean bias ignore the general trend, we also provide the Bland-Altman plot with the trend incorporated (Fig 4). At 70 bpm the rPPG under-estimated by just 5 bpm compared to the mean, but with very wide limits of agreement from -18 to 28 bpm. At 80 bpm the rPPG under-estimated by 17 bpm, again with wide limits of agreement from -9 to 43 bpm.

## Discussion

The aim of the present study was to assess in a controlled experimental setting the accuracy of the Facereader rPPG for remote HR monitoring with respect to a gold standard electrocardiograph. Although, recent evidence suggests that reliable HR measurement can be achieved using different rPPG algorithms [25, 49, 51, 55, 56], the need of further validations and cross comparisons is crucial.

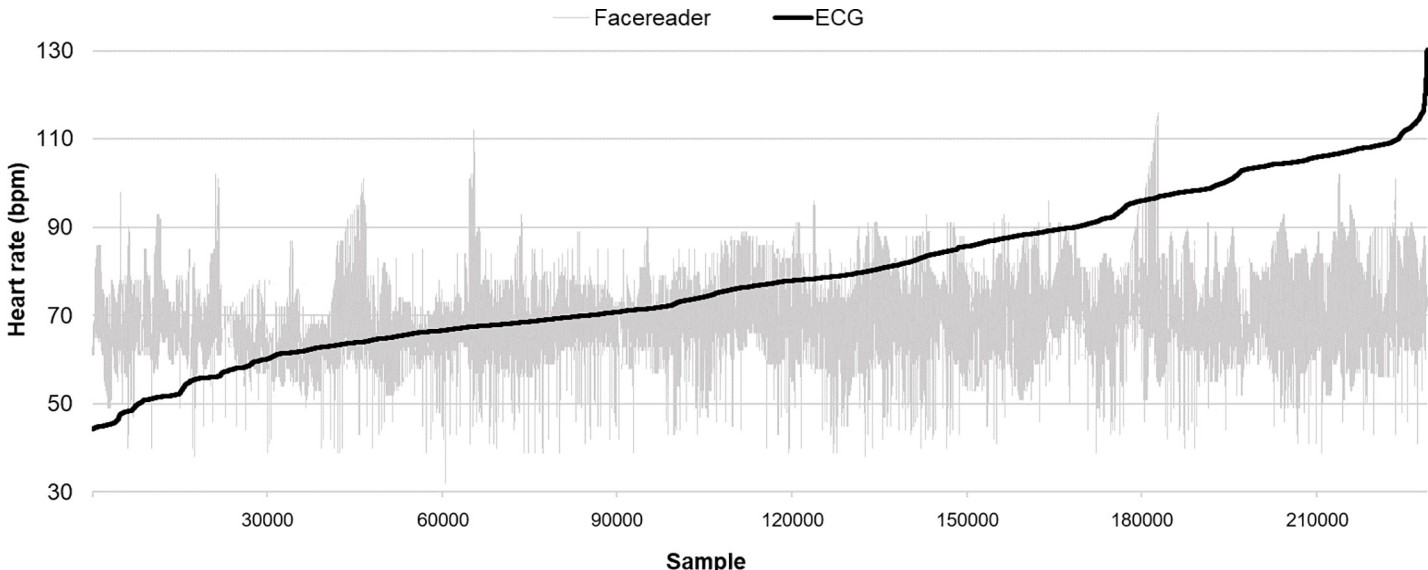

**Fig 2. Ordered HR data (Facereader rPPG vs. ECG).** Data have been ordered according to the frequencies collected by the criterion measure (ECG). (n = 230400).

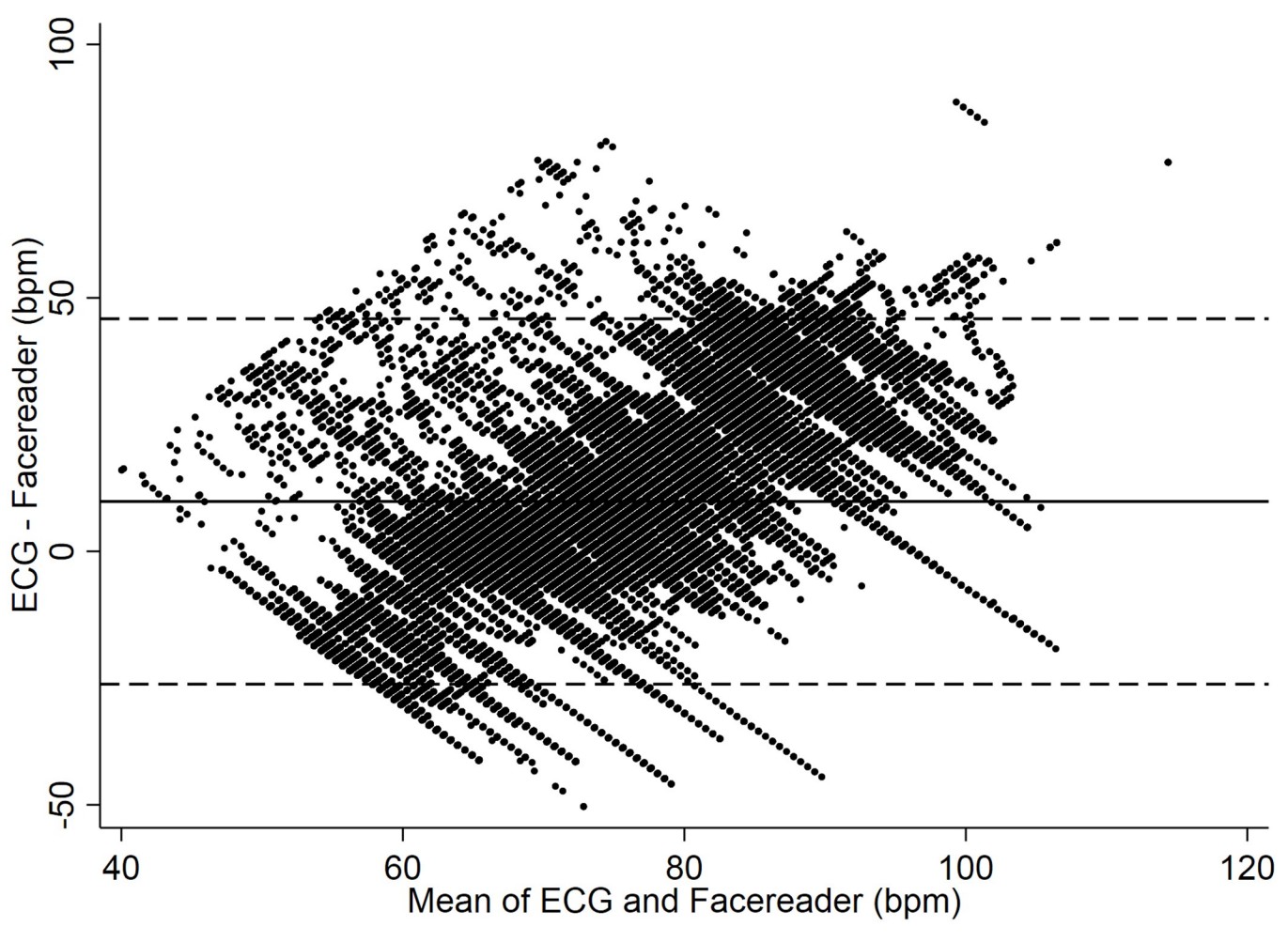

**Fig 3. HR data (Facereader vs. ECG).** Bland-Altman Plot indicating mean difference in HR detection between the Facereader rPPG and ECG criterion measure.

To the authors' knowledge the rPPG by Noldus is the only consumer product available on the market and up until today just one study involving this specific tool has been carried out [54]. Our results show that the agreement between the Facereader rPPG and the ECG is poor. The Facereader tends to over-estimate lower heart rates, and under-estimates higher heart rates compared to the ECG and the error ranges from almost -30 up to +50 bpm (see Fig 2). A first and single validation of the Facereader rPPG has been carried out by the inventors of this patented technology [53, 60]. In their study, the estimated rPPG signal was compared to a ground truth contact PPG sensor and the results of the objective performance tests show strong correlation between the estimated remote and the reference HR [60]. However, the validation was carried out using an unspecified PPG sensor, which cannot be considered a gold standard, no information was provided on the exact value of the correlation and no ICC was calculated. These issues limit our ability to compare our results with theirs. When comparing our results with those of two other related studies, it is possible to observe that in Gonzalez Viejo et al. [54] no correlation was found between the HR results of the oscillometric monitor and those obtained by analysing the video with the Facereader; and that Tasli HE et al. [60],

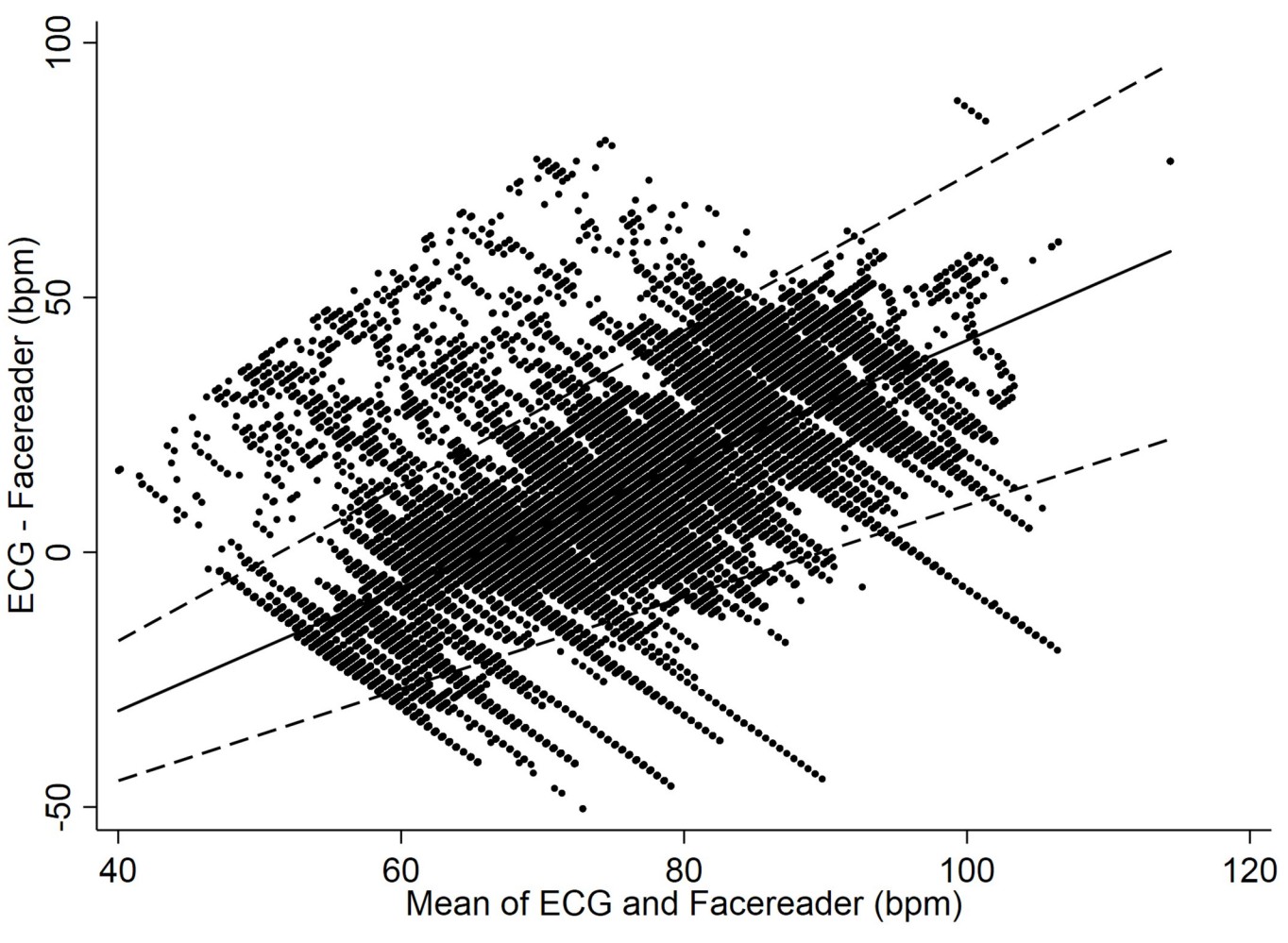

**Fig 4. HR data with trend (Facereader vs. ECG).** Bland-Altman Plot modelling a trend over continuous heart rate indicating mean difference in HR detection between the Facereader rPPG and ECG criterion measure.

which proposed a novel signal processing approach to extract the periodic component of the raw colour signal for the heart rate and variation estimation, found that the higher HR values (> 100 bpm) are underestimated by Facereader (Fig 3, points outside the confidence limit).

The lack of accurate and precise estimates from the rPPG monitor are in line with our expectations; the technology employed by the Facereader is still in its infancy and other improvements are needed to increase the precision and accuracy of this tool. The Facereader technology and the underlying technique do not overcome the limits of PPG methods in terms of accuracy of HR measurements [17].

With respect to validation studies involving other rPPG methods and ECG criterion measures, the literature is very dense, and the general idea is that rPPG seems to perform quite well. Our results are in contrast with these recent studies in which HR was reliably estimated in various scenarios through rPPG when compared to an ECG reference signal. For example, van Gastel et al. [64] found that, when rPPG was employed for estimate the cardiac activity of infants, the pulse rate can be detected with an average error which ranges from 1.5 to 2.1 bpm and overall the correct HR is detected for 87% of the time; also Fukunishi et al. [65] performed

an experiment to measure participants at rest and under cognitive stress in which the remote measurement of HR (rPPG) showed an high correlation with the ECG (around 99% accuracy). In contrast with previous investigations [50, 63, 64], our study has a large sample size (24 participants) and dataset (230400 samples): these elements together are important factors of study validity.

In general, technical features of the camera, body and head movements and ambient lighting are the main causes for the inaccuracy of any rPPG acquisition. Compatibly with the guidelines suggested by the FaceReader Reference Manual about lighting (light diffuse, no strong shadows on the face, preferably from a frontal direction), the camera and its setting (which should be able to capture a frontal view of the subject's face throughout the session with a recommended video resolution: 1280 x 720; frame rate: at least 15 fps, preferred 30 fps; distance between camera and subject: 0.5–1 m) and the skin tones of the participants, our experimental design was specifically built to control all these factors which if not respected would constitute limits to the accuracy of an optimal rPPG measurement. The fact the ICC, which indicates how strongly units in the same group resemble each other, is good (0.75), depends on the way the experimental procedure was conceived: all the participants performed the same task in the same order, under the same experimental conditions, which included stable and constant environmental lighting, temperature and minimization of head and body movements.

In case other commercial rPPG will be developed, future work will be devoted to a more naturalistic assessment and validation of these devices. Our suggestion for future research is to deepen the study of consumer products that can be used without the need for special expertise or sophisticated software and hardware. In fact, our study did not aim to assess the effectiveness and the accuracy of under development rPPG algorithms and methodologies. We aimed to evaluate the precision and accuracy of a consumer product, easily accessible by final users and industrial partners interested in monitoring HR. In this respect, our further studies will explore remote psychophysiological monitoring technologies and, considering its main imaging acquisition issues will include assessment of environmental lighting conditions, correlation with head and body movements, as well as widen the sample and testing conditions. This will allow to evaluate potential variability of the instrument performances with respect to different users' characteristics (e.g. skin type, ages) and different application scenarios (e.g. clinical settings, fitness environments, driving conditions, working environments).

## Conclusion

The Facereader[TM] rPPG allows for remote HR measurement through a video camera. Although the Facereader™ rPPG's algorithm does not represent the state-of-the-art, our assessment revealed that the agreement between the Facereader rPPG and the ECG is poor, with a mean bias of 9.8 bpm compared to the ECG gold standard. The mean bias is highly influenced by the fact that the Facereader tends to over-estimate lower heart rates, and under-estimates higher heart rates compared to the ECG. The error ranges from almost -30 up to +50 bpm. The infancy of this peculiar technology may potentially explain these results. Future investigations will further allow improvement and diffusion of this kind of technology, which has the potential to assess and monitor the personal psychophysical status in a simple, convenient and non-invasive way with important applications beyond clinical patients' monitoring [66], such as consumer analysis, e-commerce, personal fitness, driving conditions, telemedicine, and customer neuroscience.

## Supporting information

**S1 Dataset. Study data.**
(XLSX)

## Acknowledgments

We would like to thank Federica Bomben for her precious help with figures editing, as well as the anonymous reviewers for their constructive and useful comments.

## Author Contributions

**Conceptualization:** Simone Benedetto, Christian Caldato, Nicola Bartoli, Virginia Pensabene, Paolo Actis.

**Data curation:** Christian Caldato, Darren C. Greenwood, Nicola Bartoli.

**Formal analysis:** Simone Benedetto, Christian Caldato, Darren C. Greenwood, Nicola Bartoli.

**Investigation:** Simone Benedetto.

**Methodology:** Simone Benedetto, Darren C. Greenwood, Nicola Bartoli.

**Project administration:** Simone Benedetto.

**Software:** Darren C. Greenwood, Nicola Bartoli.

**Supervision:** Simone Benedetto, Paolo Actis.

**Validation:** Simone Benedetto, Darren C. Greenwood.

**Writing – original draft:** Simone Benedetto, Christian Caldato, Darren C. Greenwood, Nicola Bartoli, Virginia Pensabene, Paolo Actis.

**Writing – review & editing:** Simone Benedetto, Christian Caldato, Darren C. Greenwood, Nicola Bartoli, Virginia Pensabene, Paolo Actis.

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
