## [Decision Letter · Decision Letter 0]

12 Aug 2019

PONE-D-19-16489

Remote heart rate monitoring - Assessment of the FacereaderTM rPPg by Noldus

PLOS ONE

Dear Simone Benedetto,

Thank you for submitting your manuscript to PLOS ONE. After careful consideration, we feel that it has merit but does not fully meet PLOS ONE’s publication criteria as it currently stands. Therefore, we invite you to submit a revised version of the manuscript that addresses the points raised during the review process.

We would appreciate receiving your revised manuscript by Sep 08 2019 11:59PM. To enhance the reproducibility of your results, we recommend that if applicable you deposit your laboratory protocols in protocols.io, where a protocol can be assigned its own identifier (DOI) such that it can be cited independently in the future. For instructions see: http://journals.plos.org/plosone/s/submission-guidelines#loc-laboratory-protocols

We look forward to receiving your revised manuscript.

Kind regards,

Wajid Mumtaz

Academic Editor

PLOS ONE

Journal Requirements:

1. In your manuscript, "Caucasian" should be changed to “white” or “of [Western] European descent” (as appropriate).

2. In the manuscript and in the online submission form, please clarify whether the affiliation with TSW-XP LAB constitutes a conflict of interest.

3. Thank you for submitting the above manuscript to PLOS ONE. During our internal evaluation of the manuscript, we found significant text overlap between your submission and the following previously published works:

https://doi.org/10.1109/EMBC.2012.6346371

https://doi.org/10.1007/s11704-016-6243-6

https://doi.org/10.1109/EMBC.2015.7319857

We would like to make you aware that copying extracts from previous publications, especially outside the methods section, word-for-word is not acceptable. In addition, the reproduction of text from published reports has implications for the copyright that may apply to the publications.

Please revise the manuscript to rephrase the duplicated text, cite your sources, and provide details as to how the current manuscript advances on previous work. Please note that further consideration is dependent on the submission of a manuscript that addresses these concerns about the overlap in text with published work.

Reviewers' comments:

Reviewer's Responses to Questions

**Comments to the Author**

1. Is the manuscript technically sound, and do the data support the conclusions?

Reviewer #1: Yes

Reviewer #2: Partly

Reviewer #3: Partly

2. Has the statistical analysis been performed appropriately and rigorously? 

Reviewer #1: Yes

Reviewer #2: Yes

Reviewer #3: Yes

3. Have the authors made all data underlying the findings in their manuscript fully available?

Reviewer #1: Yes

Reviewer #2: No

Reviewer #3: Yes

4. Is the manuscript presented in an intelligible fashion and written in standard English?

Reviewer #1: Yes

Reviewer #2: Yes

Reviewer #3: Yes

5. Review Comments to the Author

Reviewer #1: This article on evaluating a commercial rPPG solution is well-written and researched in its introduction of rPPG, and the method and presentation of the study look good to me.

The authors state that their goal is only to review end-user rPPG products, of whom there are not many since rPPG is still in its early development. This somewhat lowers the contribution the article could make to the rPPG research community.

Major comments

The main issue I see with this article is that most state-of-the-art algorithms, for which such an independent evaluation would be of the major interest, are not in use commercially.

Although the authors clearly state that they want to review only commercially available algorithms, there should be greater emphasis (i.e., abstract/conclusion) that the reviewed algorithm does NOT represent the state-of-the-art.

On this note, a commercial product that is probably more advanced (and not mentioned here) is the VitalSigns Camera by Philips (http://www.ip.philips.com/licensing/program/115), but I am not sure how easy it is to get access. This needs to be reflected in the article.

Another question that I had to ask myself is how this paper could help rPPG development going forward. In my opinion, open-sourcing the dataset (of whom there are not many) for evaluation of any rPPG algorithm would be a bigger contribution than this "one off" evaluation of an outdated algorithm.

Minor comments

l. 31 This statement is only acceptable of the VitalSigns Camera by Philips is not classified as a consumer product. This should be clarified (outside the abstract).

l. 39 After a quick look at the information available, it does not seem that the Noldus FaceReader uses "recently developed" rPPG technology. The papers cited by Noldus are as old as 2014. I would drop the words "recently developed".

l. 87 There is indeed a growing interest in affect detection, but a citation is missing. The recent review "Deep Learning for Human Affect Recognition: Insights and New Developments" to be published in IEEE Transactions on Affective Computing could be suitable: https://ieeexplore.ieee.org/abstract/document/8598999

l. 139 When talking about the state-of-the-art, the paper should mention "DeepPhys: Video-Based Physiological Measurement Using Convolutional Attention Networks" by Weixuan Chen and Daniel McDuff (published at ECCV 2018), which is the most advanced approach in rPPG that I am aware of.

l. 150 The last two sentences of this paragraph lead me to expect a list of areas or methods? This could be rephrased.

l. 156 This could be a point to mention the product by Philips.

l. 228 Typo: "may causes artefacts" => "may cause artefacts"

l. 293 Again, this phrasing may have to be changed since there is the product by Philips.

l. 303 That is exactly the problem in rPPG - most studies are evaluating on their own private databases which are not comparable. Researchers should benchmark on publicly available datasets (e.g., MAHNOB-HCI) or publish their own. Why not publish this dataset?

l. 320 "larger"

l. 320 Again same point from l. 303.

l. 322 This statement of which cases are the "most plausible" should rephrased or backed up somehow.

l. 329 This is confusing: How can other commercial rPPG be evaluated if they don't exist?

Reviewer #2: The paper presents an assessment of FaceReader by Noldus, a product developed to measure remote photoplethysmographic signals and estimate pulse rate.

To be more useful to the authors, my background is as a researcher in biomedical signal and image processing. One aspect of my research consists in sensing and estimating physiological parameters from video recordings by the analysis of remote photoplethysmographic signals (which is well-correlated with the topic of the paper they submit).

I hope that these comments will help the authors to improve their paper.

1. General comments:

Introduction is in my opinion too long and can be shortened. E.g. from L. 139 to 152: the authors have specifically chosen to present two techniques (L. 139 to 152): it is not exhaustive and not particularly correlated with the main goal of this study (assessing the potential and limits of FaceReader and not presenting a signal, image or deep learning method). These parts can therefore be removed.

A section or a dedicated part in the discussion that presents comparison of the results with those from related studies (ref. [67] and [69], the latter being the original contribution) must be added.

- From [67], page 15: “When comparing the HR results from the oscillometric monitor with those obtained by analyzing the videos through FaceReader™ there was no correlation between the two methods”.

- From [69], figure 3: high HR values (>100 bpm) are underestimated (points outside the confidence limits).

Performances of FaceReader: what is the impact of subject-experiment-hardware specifications like image resolution, sensor quality (quantum efficiency), distance camera-subject, skin tone? A limitation section or a complete paragraph must be added in the discussion. A Bland-Altman for each skin tone would have been of great interest because the rPPG signal to noise ratio tends to decrease with darker skin colors. The same remark goes for motion (we could expect a worse agreement).

L. 350 to 352: these conclusions cannot be supported by the experiments (illumination is constant and artifacts due to motion have been removed).

Data: could the authors provide the data used to compute the results and, at least, some excerpts (e.g. frames from some participants)? It seems that no link(s) or archive(s) were provided.

2. Specific comments:

References: the paper is not a survey and, in my opinion, employs too many references. In addition, some references are sometimes not well chosen or are presented in too large groups (5 to 10). I would then suggest reducing this number by dropping some unrelated references. See for example L. 64, 77, 78, 87, 105 (ref. 45 to 47 are no related to augmented reality, the authors should instead cite 64), 118, 159.

On the other side, some parts need additional references:

- L. 85: “consumer neuroscience methods”

- From L. 90 to 98

- L. 128 (“Nevertheless, this method provides highly usable and accessible daily health monitoring and it is recognized to be more robust to motion artifacts if compared to infrared rPPG”)

- L. 147, ref. [64]: a better reference can be selected

- L. 177: “The ProComp Infiniti […] constitutes a gold standard for the measurement of physiological signals”

- L. 181: Einthoven triangle

L. 118 to 123: “NIR cameras allow a deeper estimate of HR” what “deeper” means in this context? I found this paragraph unclear.

L. 124: main limitations: I would recommend adding, in complement to illumination considerations, that motion can drastically affect PPG signals by engendering strong artifacts.

L. 130: in reference [61], the authors studied contact PPG signals. I am not sure that their conclusions can be directly transposed to remote PPG.

L. 133: Takano et al. [87] proposed a method to detect rPPG and estimate pulse rate in 2007, before [50].

L. 140 to 142 are misleading: the CNN is a deep learning model that detect skin pixels in a frame. rPPG is subsequently computed on these pixels of interest.

L. 194 to 196: the experiments are in fact very controlled (lab conditions, no motion), which contrast with some statements from the introduction, in particular in L. 138 or from L. 160 to 167.

Go/NoGo: I would recommend adding some details about the task.

Artefact removal (L. 229): how? Manually?

I believe that other commercial solutions like FaceReader are available, in particular on mobile devices (Android, iOS). After a quick search on Google, I also found i-virtual (http://www.i-virtual.fr/cardiasens.html) which proposes “Cardiasens”, a product apparently similar to FaceReader (the site is in French).

3. Minor corrections

- Introduction: I would recommend removing the quote marks.

- Suggestion for L. 63: mental health patients -> patients with mental disorders (it is a suggestion)

- Suggestion: imaging PPG (iPPG) instead of rPPG (remote can also correspond to the measurement of PPG signals at a distance using LEDs, photodetectors and optical components).

- Suggestion: I recommend the use of pulse rate instead of heart rate throughout the entire article (heart rate being more employed for the ECG). PPG -> pulse rate, ECG -> heart rate.

- Abbreviations that could be removed: VCG, EM, CNN.

- Abbreviations that must be defined: CI (confidence interval).

- L. 123: Kado and colleagues -> Wang and colleagues

- Suggestion: the organization of the paper can be presented at the end of the introduction.

- L. 181: recorder -> recorded

- Format of references L. 301 and 308 (Tasli et al., Benedetto et al.)

Reviewer #3: The authors compared the digital camera-based Facereader software’s ability to measure heart rate to heart rate as measured by ECG signal and found that Facereader performs poorly at (relative, but completely physiological) extremes of heart rate with error ranging from -30 to +50 bpm (mean error 9.8 bpm).

1. The authors have carefully designed their study in a way to be easily reproducible, with clear diagrams as to its setup.

2. The background provided is very detailed, with a clear overview of the underlying technologies.

3. This work aligns well with the current efforts on measuring biomarkers using the new technology. The interest in this area is growing at a fast pace and there is a need for measuring such markers as accurately as possible with as less distraction to the user as possible.

However, a few points remain that are a bit unclear in analysis:

1. It seems potentially unfair to compare Facereader only to ECG signal. Why not also compare with on-finger PPG, as in the original Facereader validation study? PPG is more commonly used for heart rate measurement than ECG is and is generally accepted as being valid. So why did authors choose not to do this?

2. They mention that their test participants were exclusively Caucasian and that this was a study limitation, but don’t touch more on why their study was designed in this manner. This point appears to have been raised by previous Reviewer #1, but the authors did not address this in detail beyond citing it as a study limitation.

3. Study participants were limited to those without neurological or cognitive disorders. What about cardiac disorders? For instance, if any participant had some type of arrhythmia, such as AFib or PVCs, those could reasonably throw off any heart rate calibration.

4. More background about the method and the basic statistics collected could have been useful. In particular, it would be useful to include more detail about the Go/No-Go task, and to include metrics such as mean/max/min for each of rest period 1, stress test 1, rest period 2, and stress test 2. It would also be useful to see the mean difference between each transition. This data was requested by Reviewer #1, but the authors did not agree with this request.

5. On a more minor note - there were a number of grammatical errors throughout and it would be good to have an external reviewer edit the paper for quality of presentation.

6. PLOS authors have the option to publish the peer review history of their article (what does this mean?). If published, this will include your full peer review and any attached files.

Reviewer #1: No

Reviewer #2: No

Reviewer #3: No

---

## [Author Response · Author response to Decision Letter 0]

30 Sep 2019

Response to reviewers

We would like to thank the reviewers for the time they spent in critically reading our manuscript. We deeply appreciate their feedback and we have addressed their comments/remarks below:

Journal Requirements:

1. In your manuscript, "Caucasian" should be changed to “white” or “of [Western] European descent” (as appropriate).

Authors: We changed “Caucasian” to Western European descendent.

2. In the manuscript and in the online submission form, please clarify whether the affiliation with TSW-XP LAB constitutes a conflict of interest.

Authors: We added the required information. Here below the details.

Funding: This work was supported by TSW XP Lab, which only provided financial support in the form of authors’ salaries [SB, CC, NB] and/or research materials. The funders had no role in study design, data collection and analysis, decision to publish, or preparation of the manuscript. The specific roles of these authors are articulated in the “Author Contributions” section.

Competing interests: The authors declare that the funding organization (TSW XP Lab) only provided financial support in the form of authors’ salaries [SB, CC, NB] and/or research materials, and did not play a role in the study design, data collection and analysis, decision to publish, or preparation of the manuscript. The specific roles of these authors are articulated in the “Author Contribution” section. The authors also confirm that this commercial affiliation does not alter their adherence to all PLOS ONE policies on sharing data and materials.

3. Thank you for submitting the above manuscript to PLOS ONE. During our internal evaluation of the manuscript, we found significant text overlap between your submission and the following previously published works:

https://doi.org/10.1109/EMBC.2012.6346371

https://doi.org/10.1007/s11704-016-6243-6

https://doi.org/10.1109/EMBC.2015.7319857

We would like to make you aware that copying extracts from previous publications, especially outside the methods section, word-for-word is not acceptable. In addition, the reproduction of text from published reports has implications for the copyright that may apply to the publications.

Please revise the manuscript to rephrase the duplicated text, cite your sources, and provide details as to how the current manuscript advances on previous work. Please note that further consideration is dependent on the submission of a manuscript that addresses these concerns about the overlap in text with published work.

Authors: We apologize for any overlap in text. We have not copied from these three reviews, but instead may have referred to the same sources at they have. In each case, we have carefully cited in our manuscript those original older sources we have in common with the reviews you list. However, to avoid re-using the same wording as the original publications, we rephrased the relevant text in our manuscript.

Reviewers' comments:

Reviewer #1: The main issue I see with this article is that most state-of-the-art algorithms, for which such an independent evaluation would be of the major interest, are not in use commercially. Although the authors clearly state that they want to review only commercially available algorithms, there should be greater emphasis (i.e., abstract/conclusion) that the reviewed algorithm does NOT represent the state-of-the-art.

Authors: We have given more emphasis to the fact that the validated algorithm does not represent the state of the art by further discussing this point both in the conclusions.

Reviewer #1: On this note, a commercial product that is probably more advanced (and not mentioned here) is the VitalSigns Camera by Philips (http://www.ip.philips.com/licensing/program/115), but I am not sure how easy it is to get access. This needs to be reflected in the article

Authors: Philips VitalSigns Camera is not a commercial product, but rather a technology available for licensing to 3rd party manufacturers. We have nevertheless introduced a reflection on Philips' product in the introduction of the paper.

Reviewer #1: Another question that I had to ask myself is how this paper could help rPPG development going forward. In my opinion, open-sourcing the dataset (of whom there are not many) for evaluation of any rPPG algorithm would be a bigger contribution than this "one off" evaluation of an outdated algorithm.

Authors: We agree with the reviewer: providing an open-source dataset would be useful, though we would wish to avoid over-fitting of algorithms to any single dataset. However, in our specific case, none of the participants gave their written consent to publish a video that depicts their face. We will consider seeking consent to open-source the datasets in our future investigations.

Reviewer #1: l. 31 This statement is only acceptable of the VitalSigns Camera by Philips is not classified as a consumer product. This should be clarified (outside the abstract).

Authors: As stated before, the product by Philips is not a consumer product.

Reviewer #1: l. 39 After a quick look at the information available, it does not seem that the Noldus FaceReader uses "recently developed" rPPG technology. The papers cited by Noldus are as old as 2014. I would drop the words "recently developed".

Authors: We agree with the reviewer. We removed the words “recently developed”.

Reviewer #1: l. 87 There is indeed a growing interest in affect detection, but a citation is missing. The recent review "Deep Learning for Human Affect Recognition: Insights and New Developments" to be published in IEEE Transactions on Affective Computing could be suitable: https://ieeexplore.ieee.org/abstract/document/8598999

Authors: This is definitely a very nice piece of work. We added this citation.

Reviewer #1: l. 139 When talking about the state-of-the-art, the paper should mention "DeepPhys: Video-Based Physiological Measurement Using Convolutional Attention Networks" by Weixuan Chen and Daniel McDuff (published at ECCV 2018), which is the most advanced approach in rPPG that I am aware of.

Authors: Thanks for the suggestion. We have added this citation.

Reviewer #1: l. 150 The last two sentences of this paragraph lead me to expect a list of areas or methods? This could be rephrased.

Authors: We agreed with Reviewer #2 “introduction is in my opinion too long and can be shortened from L. 139 to 152”. We have therefore deleted this part.

Reviewer #1: l. 156 This could be a point to mention the product by Philips.

Authors: We have introduced here a reflection on Philips' product.

Reviewer #1: l. 293 Again, this phrasing may have to be changed since there is the product by Philips.

Authors: As state above, although Philips VitalSigns Camera is not a commercial product, but rather a technology available for licensing to 3rd party manufacturers, we have introduced a reflection on Philips' product in the introduction of the paper, but not in the conclusion.

Reviewer #1: l. 303 That is exactly the problem in rPPG - most studies are evaluating on their own private databases which are not comparable. Researchers should benchmark on publicly available datasets (e.g., MAHNOB-HCI) or publish their own. Why not publish this dataset?

Authors: As stated before, we agree with the reviewer that the need of open-source dataset would be useful. However, in this specific case, none of the participants gave their written consent to publish a video that depicts their face. 

Reviewer #1: l. 322 This statement of which cases are the "most plausible" should rephrased or backed up somehow.

Authors: The sentence has been rephrased.

Reviewer #1: l. 329 This is confusing: How can other commercial rPPG be evaluated if they don't exist?

Authors: We apologize for the confusing statement and we have edited the paragraph for clarity.

Reviewer #2: Introduction is in my opinion too long and can be shortened. E.g. from L. 139 to 152: the authors have specifically chosen to present two techniques (L. 139 to 152): it is not exhaustive and not particularly correlated with the main goal of this study (assessing the potential and limits of FaceReader and not presenting a signal, image or deep learning method). These parts can therefore be removed.

Authors: We agreed with Reviewer #2. This part (i.e. L.139 to 152) has been removed.

Reviewer #2: A section or a dedicated part in the discussion that presents comparison of the results with those from related studies (ref. [67] and [69], the latter being the original contribution) must be added.

- From [67], page 15: “When comparing the HR results from the oscillometric monitor with those obtained by analyzing the videos through FaceReader™ there was no correlation between the two methods”.

- From [69], figure 3: high HR values (>100 bpm) are underestimated (points outside the confidence limits).

Authors: We agree with the reviewer and we have carefully added a part in the discussion section where the results of the two related studies are presented and compared with ours.

Reviewer #2: Performances of FaceReader: what is the impact of subject-experiment-hardware specifications like image resolution, sensor quality (quantum efficiency), distance camera-subject, skin tone? A limitation section or a complete paragraph must be added in the discussion. 

Authors: We agree with reviewer and we included a section on the Facereader rPPG performances issues in the discussion. 

Reviewer #2: A Bland-Altman for each skin tone would have been of great interest because the rPPG signal to noise ratio tends to decrease with darker skin colors. The same remark goes for motion (we could expect a worse agreement). 

Data: could the authors provide the data used to compute the results and, at least, some excerpts (e.g. frames from some participants)? It seems that no link(s) or archive(s) were provided.

Authors: Our sample was constituted by Western European descents (white skin). Therefore, we did not measure skin colour differences between our participants. As to the data, all data are already fully available without restriction in the original version of the manuscript.

Reviewer #2: L. 350 to 352: these conclusions cannot be supported by the experiments (illumination is constant and artifacts due to motion have been removed).

Authors: We apologize for the confusing statement and we have edited the paragraph for clarity.

Reviewer #2: References: the paper is not a survey and, in my opinion, employs too many references. In addition, some references are sometimes not well chosen or are presented in too large groups (5 to 10). I would then suggest reducing this number by dropping some unrelated references. See for example L. 64, 77, 78, 87, 105 (ref. 45 to 47 are no related to augmented reality, the authors should instead cite 64), 118, 159.

Authors: We agree with the reviewer. We reduced the number of references by dropping the unrelated ones.

Reviewer #2: On the other side, some parts need additional references:

- L. 85: “consumer neuroscience methods”

- From L. 90 to 98

- L. 177: “The ProComp Infiniti […] constitutes a gold standard for the measurement of physiological signals”

- L. 181: Einthoven triangle

Authors: We added citations for all of them. As to the Procomp Infiniti, the device is a professional tool used in the clinical and experimental field (for example in biofeedback training and therapy) and constitutes a gold-standard for the measurement of ECG signal. It has been employed in more than 300 (published) experimental studies, including our previous study on the assessment of the Fitbit Charge 2 for monitoring heart rate, published in PlosOne one year ago. (Benedetto, S., Caldato, C., Bazzan, E., Greenwood, D. C., Pensabene, V., & Actis, P. (2018). Assessment of the Fitbit Charge 2 for monitoring heart rate. PloS one, 13(2), e0192691).

Reviewer #2: L. 128 (“Nevertheless, this method provides highly usable and accessible daily health monitoring and it is recognized to be more robust to motion artifacts if compared to infrared rPPG”)

Authors: The sentence has been removed.

Reviewer #2: L. 147, ref. [64]: a better reference can be selected

Authors: This part (i.e. L.139 to 152) has been removed. 

Reviewer #2: L. 118 to 123: “NIR cameras allow a deeper estimate of HR” what “deeper” means in this context? I found this paragraph unclear.

Authors: We apologize for this. The sentence has been rephrased.

Reviewer #2: L. 124: main limitations: I would recommend adding, in complement to illumination considerations, that motion can drastically affect PPG signals by engendering strong artifacts.

Authors: We agree with the reviewer. We added the suggested sentence.

Reviewer #2: L. 130: in reference [61], the authors studied contact PPG signals. I am not sure that their conclusions can be directly transposed to remote PPG.

Authors: The sentence has been removed.

Reviewer #2: L. 133: Takano et al. [87] proposed a method to detect rPPG and estimate pulse rate in 2007, before [50]. 

Authors: We apologize for this. The sentence has been rephrased.

Reviewer #2: L. 140 to 142 are misleading: the CNN is a deep learning model that detect skin pixels in a frame. rPPG is subsequently computed on these pixels of interest.

Authors: This part (i.e. L.139 to 152) has been removed.

Reviewer #2: L. 194 to 196: the experiments are in fact very controlled (lab conditions, no motion), which contrast with some statements from the introduction, in particular in L. 138 or from L. 160 to 167.

Authors: We apologize for this. The sentence has been rephrased.

Reviewer #2: Go/NoGo: I would recommend adding some details about the task.

Authors: We added more details regarding the task.

Reviewer #2: Artefact removal (L. 229): how? Manually?

Authors: Yes, manually. We added this detail in the revised manuscript.

Reviewer #2: I believe that other commercial solutions like FaceReader are available, in particular on mobile devices (Android, iOS). After a quick search on Google, I also found i-virtual (http://www.i-virtual.fr/cardiasens.html) which proposes “Cardiasens”, a product apparently similar to FaceReader (the site is in French).

Authors: The solutions available on mobile devices were excluded because they are closer to game/fun applications rather than real commercial ones. For this reason, we did not consider them as reliable benchmark. As to Cardiasens, we contacted several times the company but did not receive any reply. We also verified if any patent or publication was referred to Cardiasens but nothing was found.

Reviewer #2: 

- Introduction: I would recommend removing the quote marks.

- Suggestion for L. 63: mental health patients -> patients with mental disorders (it is a suggestion)

- Abbreviations that could be removed: VCG, EM, CNN.

- Abbreviations that must be defined: CI (confidence interval).

- L. 123: Kado and colleagues -> Wang and colleagues

- L. 181: recorder -> recorded

- Format of references L. 301 and 308 (Tasli et al., Benedetto et al.)

Authors: These issues were addressed in the revised manuscript.

Reviewer #2: 

- Suggestion: imaging PPG (iPPG) instead of rPPG (remote can also correspond to the measurement of PPG signals at a distance using LEDs, photodetectors and optical components).

- Suggestion: I recommend the use of pulse rate instead of heart rate throughout the entire article (heart rate being more employed for the ECG). PPG -> pulse rate, ECG -> heart rate.

- Suggestion: the organization of the paper can be presented at the end of the introduction.

Authors: We thank the reviewer for these suggestions, but we would prefer to keep the text as it is. As to the employment of iPPG in lieu of rPPG, we do not think it is correct. Noldus infact refers to rPPG and to heart rate for its patented technology (see https://www.noldus.com/facereader/remote-photoplethysmography-facereader). Furthermore, the large majority of the literature refers to rPPG. As to the suggestion regarding the organization of the paper at the end of the introduction, we do not think it is necessary. 

Reviewer #3: 1. It seems potentially unfair to compare Facereader only to ECG signal. Why not also compare with on-finger PPG, as in the original Facereader validation study? PPG is more commonly used for heart rate measurement than ECG is and is generally accepted as being valid. So why did authors choose not to do this?

Authors: We decided to employ an ECG signal, just because the ECG is the reference method for this kind of assessment. Unfortunately, the on-finger PPG cannot be considered a gold standard, and therefore cannot be employed in any (official) validation study. If we used anything other than the ECG, we would not know if disagreements were because of the Facereader or the reference measure.

Reviewer #3: 2. They mention that their test participants were exclusively Caucasian and that this was a study limitation, but don’t touch more on why their study was designed in this manner. This point appears to have been raised by previous Reviewer #1, but the authors did not address this in detail beyond citing it as a study limitation.

Authors: We addressed this issue by reinforcing the motivations in the Materials and Methods’ section. 

Reviewer #3: 3. Study participants were limited to those without neurological or cognitive disorders. What about cardiac disorders? For instance, if any participant had some type of arrhythmia, such as AFib or PVCs, those could reasonably throw off any heart rate calibration.

Authors: None of the participants suffered from cardiac disorders. We now include this information. 

Reviewer #3: 4. More background about the method and the basic statistics collected could have been useful. In particular, it would be useful to include more detail about the Go/No-Go task, and to include metrics such as mean/max/min for each of rest period 1, stress test 1, rest period 2, and stress test 2. It would also be useful to see the mean difference between each transition. This data was requested by Reviewer #1, but the authors did not agree with this request.

Authors: We thank the reviewer for this comment. We have included more details about the Go/No-Go task in the Materials and methods section. Regarding the additional metrics required, we think that these kinds of parameters were not part of the research objectives. The objective of the study was to assess the accuracy of a consumer rPPG system with respect to HR monitoring and compare its performance to the gold standard ECG collecting a large amount of data, spanning the widest possible range of HR frequencies and we have not catalogued HR data across the stress/rest phases.

Reviewer #3: 5. On a more minor note - there were a number of grammatical errors throughout and it would be good to have an external reviewer edit the paper for quality of presentation.

Authors: We would like to thank the reviewer for pointing out this issue. We have carefully revised the manuscript.

---

## [Decision Letter · Decision Letter 1]

8 Nov 2019

Remote heart rate monitoring - Assessment of the FacereaderTM rPPg by Noldus

PONE-D-19-16489R1

Dear Dr. Simone Benedetto,

We are pleased to inform you that your manuscript has been judged scientifically suitable for publication and will be formally accepted for publication once it complies with all outstanding technical requirements.

With kind regards,

Wajid Mumtaz

Academic Editor

PLOS ONE

Additional Editor Comments (optional):

Reviewers' comments:

Reviewer's Responses to Questions

**Comments to the Author**

1. If the authors have adequately addressed your comments raised in a previous round of review and you feel that this manuscript is now acceptable for publication, you may indicate that here to bypass the “Comments to the Author” section, enter your conflict of interest statement in the “Confidential to Editor” section, and submit your "Accept" recommendation.

Reviewer #1: All comments have been addressed

Reviewer #3: All comments have been addressed

2. Is the manuscript technically sound, and do the data support the conclusions?

Reviewer #1: Yes

Reviewer #3: Partly

3. Has the statistical analysis been performed appropriately and rigorously? 

Reviewer #1: Yes

Reviewer #3: Yes

4. Have the authors made all data underlying the findings in their manuscript fully available?

Reviewer #1: Yes

Reviewer #3: (No Response)

5. Is the manuscript presented in an intelligible fashion and written in standard English?

Reviewer #1: Yes

Reviewer #3: (No Response)

6. Review Comments to the Author

Reviewer #1: I am happy with the changes made regarding my previous comments. As a result I consider the paper ready for publication, given the following minor typos are addressed:

- l. 188-189: There is an unmatched parenthesis and the sentence seems incomplete

- l. 308: There should be no comma after "Although"

Reviewer #3: I'm still not convinced about the lack of finger PPG - if you want to use ECG as a reference standard fine, but it would be helpful to also have finger PPG and be able to discuss commonly used heart rate measurements.

7. PLOS authors have the option to publish the peer review history of their article (what does this mean?). If published, this will include your full peer review and any attached files.

Reviewer #1: No

Reviewer #3: No

---

## [Editor Report · Acceptance letter]

14 Nov 2019

PONE-D-19-16489R1 

Remote heart rate monitoring - Assessment of the FacereaderTM rPPg by Noldus 

Dear Dr. Benedetto:

I am pleased to inform you that your manuscript has been deemed suitable for publication in PLOS ONE. Congratulations! Your manuscript is now with our production department. 

With kind regards,

on behalf of

Dr. Wajid Mumtaz 

Academic Editor

PLOS ONE